# Weakly Supervised Video Anomaly Detection and Localization with Spatio-Temporal Prompts

Submission Id: 433*

## ABSTRACT

Current weakly supervised video anomaly detection (WSVAD) task aims to achieve frame-level anomalous event detection with only coarse video-level annotations available. Existing works typically involve extracting global features from full-resolution video frames and training frame-level classifiers to detect anomalies in the temporal dimension. However, most anomalous events tend to occur in localized spatial regions rather than the entire video frames, which implies existing frame-level feature based works may be misled by the dominant background information and lack the interpretation of the detected anomalies. To address this dilemma, this paper introduces a novel method called **STPrompt** that learns spatio-temporal prompt embeddings for weakly supervised video anomaly detection and localization (WSVADL) based on pre-trained vision-language models (VLMs). Our proposed method employs a two-stream network structure, with one stream focusing on the temporal dimension and the other primarily on the spatial dimension. By leveraging the learned knowledge from pre-trained VLMs and incorporating natural motion priors from raw videos, our model learns prompt embeddings that are aligned with spatio-temporal regions of videos (e.g., patches of individual frames) for identify specific local regions of anomalies, enabling accurate video anomaly detection while mitigating the influence of background information. Without relying on detailed spatio-temporal annotations or auxiliary object detection/tracking, our method achieves state-of-the-art performance on three public benchmarks for the WSVADL task.

## CCS CONCEPTS

• **Computing methodologies → Scene anomaly detection**; *Visual content-based indexing and retrieval.*

## KEYWORDS

Video anomaly detection, spatio-temporal detection, language-image pre-training

## 1 INTRODUCTION

As a challenging and long-standing problem, video anomaly detection (VAD) has garnered significant attention from the computer vision community. The core objective of VAD is to detect various real-world anomalous events, holding immense potential for numerous practical applications, particularly in the realm of surveillance. For example, an intelligent video surveillance system equipped with anomaly detection capabilities can promptly perceive potential dangers, thereby facilitating timely interventions to enhance public security. Early studies have primarily focused on semi-supervised VAD [6, 11, 14, 31, 37, 47], where the task is to learn normal patterns by solely utilizing normal videos, with abnormal events identified as those deviating from the learned normal pattern. However, these

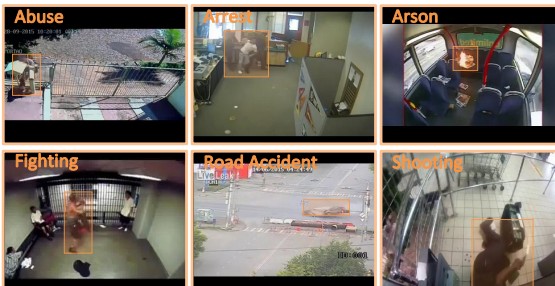

**Figure 1: Visualizations of spatial size of anomalies in surveillance videos. Video samples are taken from UCF-Crime [52].**

methods encounter limitations as they lack knowledge about abnormal videos, potentially leading to a high false alarm rate.

Weakly supervised video anomaly detection (WSVAD) has emerged as a prominent research topic in recent years, distinct from its semi-supervised counterpart in that both normal and abnormal videos are available during the training stage. The objective of WSVAD is to achieve frame-level anomaly detection with weak or coarse annotations (i.e., video-level labels). Existing works typically involve extracting features from full-resolution frames using pre-trained models such as I3D [2], Transformer [7], and CLIP [45], followed by training a classifier based on the multiple instance learning (MIL) mechanism to predict anomalous events at the frame level. While these approaches yield promising results as a standard practice, they often overlook a crucial aspect: abnormal events tend to occur in localized spatial regions rather than spanning the entire full-resolution frame, especially in surveillance scenarios. Drawing insights from the popular benchmark UCF-Crime [52], we illustrate this observation with examples depicted in Figure 1. For clarity, the anomalous regions are delineated using orange bounding boxes. It is evident that various types of anomalies manifest in diverse spatial locations and sizes; however, the spatial extent of anomalies is typically small compared to the dimensions of the full-resolution video frame. Nevertheless, existing works compress entire frames into single features, thereby neglecting crucial region-level abnormal details, and leading subsequent classifiers to rely heavily on dominant background information. Moreover, such operations further cause that the VAD model lacks reliability and interpretability, as it does not verify whether the detection align with the actual spatial location of anomalies. Consequently, certain true positive detection may be merely "lucky" coincidences of erroneous detection (such as irrelevant background or events) to the abnormal events [31].

Therefore, how to explicitly exploit the spatio-temporal fusion [15, 21], and detect anomalies from when and where perspectives, is an important direction for exploration. It is worth noting that our work is not the first attempt to address weakly supervised video anomaly detection and localization (WSVADL). Several

previous studies [28, 33, 53, 64] have also endeavored to lever-age spatio-temporal relation capabilities to enhance frame-level VAD under weak supervision. However, these approaches often necessitate complex and resource-intensive relation modeling. For example, Liu et al. [33] re-annotated the UCF-Crime benchmark, subsequently training detection models with full spatio-temporal annotations. On the other hand, Wu et al. [64] paved an another way for weakly supervised spatio-temporal VAD, which took inspirations from spatio-temporal action localization [26, 42], and used a tubelet-level detector based on pre-trained object detectors and hierarchical clustering to detect possible spatio-temporal anomalies. These methods have demonstrated improved frame-level anomaly detection results over conventional WSVAD methods by mitigating the influence of irrelevant background through intricate spatio-temporal modeling processes. Nevertheless, these solutions are either complicated, e.g., multi-scale spatial pyramid training [28] and spatio-temporal coherence modeling [53] or heavily dependent on auxiliary modules for object detection and tracking [64] and detailed spatio-temporal annotations. [33]

In this paper, enlighted by the success of large pre-trained vision-language models (VLMs) on industrial defect detection [4, 23, 81, 82], we propose a novel method **STPrompt** that learns spatio-temporal prompt embeddings built on the top of VLMs for WSVADL. Different from previous works [28, 33, 53, 64], our approach is both conceptually straightforward and practically effective. Specifically, we first segment the video into frames and further then split each frame into patches, and video anomaly detection and localization can thus be conceptualized as frame-level and patch-level classification, obviating the reliance on object detection and tracking. Concurrently, to alleviate the complexity associated with spatio-temporal relation modeling, we explicitly decompose spatio-temporal VAD into two distinct sub-tasks: temporal anomaly detection and spatial anomaly localization. For temporal anomaly detection, alongside standard temporal modeling, we introduce a simple yet effective spatial attention aggregation ($SA^2$) mechanism aimed at enhancing background denoising. This approach leverages motion priors derived from the intrinsic attributes of videos. Regarding spatial anomaly localization, we capitalize on the established image-to-concept capability of VLMs, taking a significant step forward towards training-free spatial anomaly localization without full supervision. By leveraging related concepts, we identify abnormal patches. Our approach addresses the limitations of prior works while achieving superior performance across three public benchmarks: UCF-Crime [52], ShanghaiTech [38], and UBnormal [1].

To summarize, the main contributions of this paper are threefold:
• A novel model named STPrompt is proposed to address spatio-temporal video anomaly detection under weak video-level supervisions. To our knowledge, STPrompt represents the first endeavor to efficiently transfer pre-trained vision-language knowledge from VLMs to simultaneously tackle temporal anomaly detection and spatial anomaly localization.
• To mitigate the requirement of extra auxiliary information and intricate modeling strategies, STPrompt decouples the WSVADL task into temporal anomaly detection and spatial anomaly localization. A spatial attention aggregation mechanism is devised in STPrompt to filter irrelevant background for temporal anomaly detection. Besides, a large language models (LLMs)-enabled, training-free

anomaly localization method is introduced to obtain fine-grained text prompts for spatial anomaly localization.
• Extensive experiments on three widely-used benchmarks show the superiority of STPrompt over state-of-the-art competing methods. It performs substantially better than, or on par with, the recent competing methods in anomaly detection, while largely outperforming them in TIoU for anomaly localization across all three datasets, e.g., by a margin of about 1.9% on UCF-Crime, 5.7% on ShanghaiTech, and 4.5% on UBnormal compared to VadCLIP [71].

## 2 RELATED WORK

### 2.1 Video Anomaly Detection

*2.1.1 Semi-supervised VAD.* The advent of deep learning revolutionized the field of semi-supervised VAD, with the mainstream of research focusing on convolutional neural networks (CNNs) [10, 22, 29, 35, 43, 61, 66, 69, 75], recurrent neural networks (RNNs) [51, 74], and transformers [63, 73], with many of these approaches adopting self-supervised learning principles. For example, several studies [17, 38, 77] utilize 2D-CNNs, 3D-CNNs, and RNN-based autoencoders to reconstruct normal events and identify abnormal events based on the magnitude of the reconstruction error. Liu et al. [34] proposed a CNN-based video prediction network to predict future video frames based on previous frames, while Yang et al. [73] employed transformers to extract video features and then reconstructed video events based on key-frames. Yu et al. [75] introduced a novel approach called video event completion to address gaps existing in reconstruction or frame prediction methods. Several of these approaches also address spatial anomaly localization. For instance, Li et al. [31] divided the visual field into overlapping regions and learned a global mixture model using only patches around the current frame, with regions least similar to their surroundings deemed most likely to be abnormal. Wu et al. [66] similarly divided the visual field into overlapping regions and trained a deep one-class model to discriminate abnormal regions.

*2.1.2 Weakly supervised VAD.* Weakly supervised video anomaly detection [3, 9, 33, 52, 55, 64, 76] has emerged as a prominent research focus in recent years. Sultani et al. [52] were among the pioneers, introducing a deep MIL model that treats a video as a bag and its segments as instances. By utilizing ranking loss with bag-level labels, their model aims to maximize the separation between the most anomalous instances in positive bags and negative bags. Subsequent studies have endeavored to enhance the positive design aspect of WSVAD. For instance, Zhong et al. [78] proposed a graph convolutional network (GCN)-based method to model feature similarity and temporal consistency between video segments. Tian et al. [55] devised robust temporal feature magnitude learning, significantly improving the MIL approach's robustness to negative instances from abnormal videos. Li et al. [30] and Huang et al. [19] introduced transformer-based multi-sequence learning frameworks to capture temporal relationships between frames. Zhou et al. [79] proposed dual memory units and an uncertainty learning scheme to better distinguish patterns of normality and anomaly. Wu et al. [67, 68] introduced a novel multi-modal dataset and a fine-grained weakly supervised VAD method capable of distinguishing between different types of anomalous frames. More recently, pre-trained

vision-language models have garnered significant attention in the VAD community. VadCLIP [71] was the first to efficiently transfer pre-trained language-visual knowledge from CLIP [45] to weakly supervised VAD, achieving state-of-the-art performance. Pu et al. [44] attempted to enhance WSVAD by learning prompt-enhanced context features.

## 2.2 Image Anomaly Detection with Prompts

Generally, image anomaly detection aims to localize anomalies in images like industrial defect images, predicting an image or a pixel as normal or anomalous. Typical works [5, 18, 20, 48, 54, 57, 72] mainly focused on one-class or self-supervised anomaly detection, which only requires normal images. Recently, exploiting VLMs with prompts has emerged as a successful enabler for this task, especially for the zero/few-shot setting. WinCLIP [23] introduced a language-guided paradigm for zero-shot industrial defect detection. AnomalyCLIP [81] adapted CLIP for zero-shot industrial defect detection across different domains, which learn object-agnostic text prompts that capture generic normality and abnormality. InC-TRL [82] learned residual features between query images and few-shot in-context normal images to build generalist models for image anomaly detection. These CLIP-based works inspired us to spatially local anomaly in videos, but our method is more succinct and does not require learnable parameters in the prompts.

## 3 METHOD

### 3.1 Overview

Previous WSVAD task supposes that only video-level labels are given for model training, and encourages the model to predict whether each video frame is abnormal at the test time, where the detection granularity falls into the frame level. In comparison to WSVAD, WSVADL is a more challenging task, which assumes that the model is supposed to detect anomalies at a finer level, i.e., the pixel level, while keeping the supervisions unchanged. Mathematically, given a set of training samples $\{\mathcal{V}, \mathcal{Y}_b, \mathcal{Y}_c\}$, where $\mathcal{V}$, $\mathcal{Y}_b$, and $\mathcal{Y}_c$ denote the sets of video, video-level binary label, and video-level category label, respectively. For each video sample $v$, it has two corresponding labels, namely, $y_b$ and $y_c$. Here $y_b \in \{0, 1\}$, and $y_b = 1$ indicates that $v$ includes anomalies; and $y_c \in \mathcal{R}^{1+C}$, in which $C$ is the number of abnormal categories.

As aforementioned, the main limitation of previous spatio-temporal VAD works [28, 33, 53, 64] is that they rely on labor-intensive spatio-temporal annotations, detector-dependent pre-processing, and computationally expensive spatio-temporal modeling. Compared to these works, our STPrompt is conceptually simple yet practically effective, which is demonstrated in Figure 2. To move beyond the above limitations, there are a series of dedicated designs in our STPrompt. Firstly, based on this routine operation of splitting a video into multiple frames, we further split each frame into multiple patches. Through such an operation, WSVADL can be considered as a coarse frame-level and patch-level classification task without the requirement of any detection pre-processing. On this case, a natural way is to directly treat all spatial patches as instances, and then use the MIL mechanism to predict the anomaly confidence of each patch. However, such a readily implemented way is computationally heavy and can not be easily optimized [13, 27]. Therefore,

to reduce the spatio-temporal modeling complexity and optimization difficulty, we then factor the WSVADL task into two sub-tasks, i.e., temporal anomaly detection and spatial anomaly localization. For temporal anomaly detection, we introduce a dual-branch model built on the top of CLIP, meanwhile, we design two key modules, on the one hand, a spatial attention aggregation assists the temporal detection model in focusing on potential spatial location of anomalies. On the other hand, a typical temporal adapter enhances the temporal context capture capabilities of the temporal detection model. For spatial anomaly localization, to address the challenges posed by insufficient supervisions, we design a training-free anomaly localization strategy with the basis of "image-to-concept" capacity of VLM.

### 3.2 Motion Prior-aware Spatio-Temporal Prompt Learning for Anomaly Detection

Inspired by the pioneer work VadCLIP [71], we also introduce a dual-branch framework, namely, classification branch and alignment branch. Specifically, given a video $v$, we employ a frozen image encoder of CLIP to extract the frame-level feature $x_{CLIP} \in \mathcal{R}^{T \times D}$, where $T$ is the length of video $v$, and $D$ is the feature dimension. Then these feature are fed into two branches after a series of information enhancements, classification branch is to directly predict the anomaly confidence $A \in \mathcal{R}^{T \times 1}$ by a binary classifier, another align branch is to compute the anomaly category probability $M \in \mathcal{R}^{T \times (1+C)}$ by means of the image-to-concept alignment. With $A$ and $M$ in hands, we adopt the typical TopK [67] and the recent MIL-Align [71] strategies to compute the video-level anomaly prediction and category prediction, respectively, these predictions are subsequently used to calculate losses and provide data support for model optimization. Throughout the whole process, we devise two modules to encourage the model to focus on anomalies from the spatial and temporal dimensions, which are illustrated in the following sections.

#### 3.2.1 Motion prior-aware spatial attention aggregation.
Although we explicitly disentangle WSVADL into two independent tasks, i.e., temporal anomaly detection and spatial anomaly detection, for the temporal anomaly detection task, we still require the critical spatial local anomalies as assistance information. This is because potential spatial anomalies can eliminate the noise effect caused by the irrelevant backgrounds, after all, most anomalies may occupy a small spatial region. For this problem, a majority of previous works completely ignore spatial anomaly information, and a tiny minority attempts to learn the interaction between spatial patches and video frames. The former works lacks the consideration of the use of individual spatial contents, while the latter works inevitably incurs excessive computational costs. Therefore, we propose a novel spatial attention aggregation (SA$^2$) scheme to capture key spatial information with low computational costs. As we know, the whole frames consist of the background of the scene and foreground of action, and anomalous events often occur with foreground objects, thus focusing on the spatial foreground captures potentially anomalous events. The common methods for locating the foreground include object detection algorithms [46] or optical flow [8], but these require high computation costs. Here, we propose a considerably simple and efficient method named SA$^2$, inspired by motion priors based

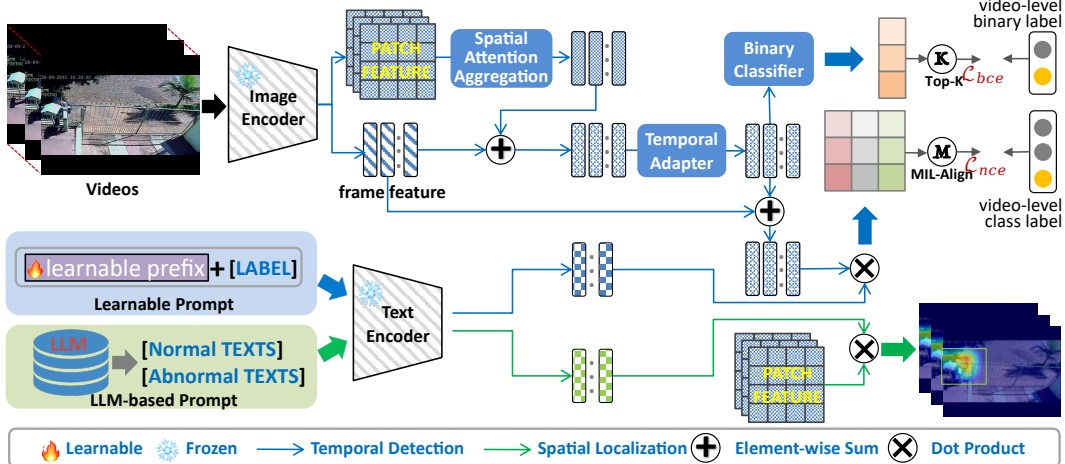

**Figure 2: The pipeline of our proposed STPrompt.**

works [59, 65]. Specifically, given the frame-level feature $x_{CLIP}$ and its corresponding spatial feature $x_{PATCH} \in \mathcal{R}^{T \times H \times W \times D}$, where $H$ and $W$ are the height and width of the spatial feature, we argue that when most abnormal events occur, the corresponding location within spatial feature would change significantly [65]. Therefore, we compute the difference between frames to obtain the motion magnitude:

$$Mo[i] = L2(x_{PATCH}[i] \times 2 - x_{PATCH}[i-1] - x_{PATCH}[i+1]), \quad (1)$$

where the size of $Mo$ is $T \times H \times W$, L2 is the L2 normalization applied in the channel dimension, and $i$ denotes the $i$-th frame. Then we use the TopK mechanism to select a fixed number of patch-level feature $x_{Mo} \in \mathcal{R}^{T \times K \times D}$ with the highest motion magnitude $Mo_{TOP} \in \mathcal{R}^{T \times K \times 1}$, where $K < H \times W$, and then compute the attention to obtain the aggregate spatial feature:

$$Attention[i] = SoftMax(Mo_{TOP}[i]), \quad (2)$$

$$x_{AS}[i] = Attention[i]^\top x_{Mo}[i]. \quad (3)$$

Different from $x_{CLIP}$ in which all pixels in each frame have nearly equal influence for anomaly detection, $x_{AS}$ places a heavy focus on potential anomaly locations. No matter how the spatial region of abnormal events changes, these two features, i.e., $x_{CLIP}$ and $x_{AS}$, can extract key abnormal information from the local and global perspectives. In other words, they are complementary.

*3.2.2 Temporal CLIP adapter.* As aforementioned, we adopt the pre-trained image encoder of CLIP to extract frame-level features, which contain momentary information but lacks a global temporal context critical for the VAD task. This motivates us to study temporal context modeling. We propose the temporal adapter that is similar to a vanilla multi-layer Transformer encoder, consisting of self-attention, layer normalization (LN), and feed-forward networks (FFN). Following [40], positional encoding is not applied. The main difference between temporal adapter and Transformer encoder lies in the self-attention, which is based on relative distance instead of feature similarity [70]. The adjacency matrix in self-attention is calculated as $Ma[i,j] = \frac{-|i-j|}{\sigma}$, where the similarity between $i$-th

and $j$-th frames only determined by their relative temporal distance. $\sigma$ is a hyper-parameter to control the range of influence of distance relation. In this work, we add $x_{CLIP}$ and $x_{AS}$ together, and feed the summation feature into temporal adapter, thus empowering CLIP with temporal modeling capability, which can be formulated as follows:

$$\begin{aligned} x_{TM} &= LN\left(SoftMax(Ma)(x_{CLIP} + x_{AS})\right), \\ x_{TA} &= LN\left(FFN(x_{TM}) + x_{TM}\right). \end{aligned} \quad (4)$$

*3.2.3 Dual-branch prompt learning.* After obtaining the deeply processed features, we require the model to predict the frame-level anomaly confidence. Due to the proven performance of VadCLIP, we further adopt its dual-branch detection framework. The one branch is a classification branch (C-Branch), a simple linear layer with the number of neuron of one, which takes $x_{TA}$ as input, and generates the anomaly confidence $A$. Another branch is an alignment branch (A-Branch), which takes video features and textual embedding of labels as input, and yields the anomaly category probability $M$. To be specific, we create the image feature by adding the original CLIP feature $x_{CLIP}$ and the output of temporal adapter $x_{TA}$ together, combining the pre-trained knowledge from CLIP and newly learned contextual information. For the textual embedding of labels, we take inspiration from CoOp [80], we add a learnable prefix prompt embedding into the category embeddings, where the category embeddings are created by transforming original text categories, e.g, *Fighting, Shooting, Car accident*, into class tokens through CLIP Tokenizer, and then put them into text encoder of CLIP. Mathematically, we concatenate the category embedding for class $i$, $t_{c_i}$, with the learnable embedding $\{e_1, ..., e_l\}$ that consists of $l$ context tokens to form a complete sentence token, and thus, the input of text encoder for one class is presented as $\{e_1, \cdots, e_l, t_{c_i}\}$. The overall label prompt embedding $Prompt \in \mathcal{R}^{(1+C) \times D}$ is the CLS token output of text encoder. With $Prompt$ and $x_{CLIP} + x_{TA}$ in hands, $M$ is generated by,

$$M = \frac{[x_{CLIP} + x_{TA}]Prompt^\top}{\|x_{CLIP} + x_{TA}\|_2 \cdot \|Prompt\|_2}. \quad (5)$$

*3.2.4 Objective function.* Following the setup of [71], TopK-based classification objective function is adopted for classification branch, which can be presented as follows,

$$p_b = Mean(TopK(A)),$$
$$\mathcal{L}_{class} = -y_b Log p_b - (1 - y_b) Log(1 - p_b), \tag{6}$$

where $TopK$ means select the set of $k$-max frame-level confidences in $A$ for the video $v$. $\mathcal{L}_{class}$ is the binary cross-entropy between $p_b$ and video-level binary labels $y_b$.

MIL-Align based objective function is used for alignment branch, which is based on anomaly category probability $M$. For each column of $M$, we select $k$-max similarities and compute the average to measure the alignment degree between $v$ and the current class. Then we can obtain a vector $S = \{s_1, ..., s_{(1+C)}\}$ that represents the similarity between $v$ and all classes. Then we compute the loss $\mathcal{L}_{align}$ as follows,

$$p_{c_i} = \frac{exp\left(s_i/\tau\right)}{\sum_j exp\left(s_j/\tau\right)}, \tag{7}$$

$$\mathcal{L}_{align} = -\sum_i^{1+C} y_{c_i} Log p_{c_i}, \tag{8}$$

where $p_{c_i}$ is the prediction of the $i$-th class, and $\tau$ refers to the temperature hyper-parameter for scaling.

To learn discriminative prompt embeddings, we also introduce a contrastive loss to make all textual embeddings more dispersible. Specifically, we calculate cosine similarity between label prompt embeddings, and compute the contrastive loss $\mathcal{L}_{const}$ as follows,

$$\mathcal{L}_{const} = \sum_i \sum_j max\left(0, \frac{Prompt_{c_i}^\top Prompt_{c_j}}{\|Prompt_{c_i}\|_2 \cdot \|Prompt_{c_j}\|_2}\right). \tag{9}$$

The final objective function is the weighted sum of the above three loss functions:

$$\mathcal{L} = \mathcal{L}_{class} + \alpha \mathcal{L}_{align} + \beta \mathcal{L}_{const}. \tag{10}$$

## 3.3 LLM-Enabled Text Prompting for Spatial Anomaly Localization

The core of this operation is that how to locate anomaly regions. Thanks to the emerging paradigm of pre-trained VLMs, we take a step forward to training-free spatial anomaly localization. Inspired by CLIP-based industrial defect detection works [23, 81, 82], we regard the spatial anomaly localization as a spatial patch retrieval process given text queries. Specifically, we suppose a test video frame is deemed as an anomaly frame due to its high anomaly score. We then obtain its patch-level feature map $x_P \in \mathcal{R}^{H \times W \times D}$ by the sliding window scheme, in which the patches are generated in the same way as $x_{PATCH}$. Here the sliding windows scheme means that we first generate a set of image patches with a fixed-size window of $P \times P$ by sliding the window with a stride of $S$, i.e., a operation similar to convolutions, and then feed these image patches into the image encoder of CLIP to obtain the corresponding embedding of the CLS token. Notably, we do not adopt the natural dense representations, i.e., the penultimate feature maps in CLIP, though its generation is simpler than the sliding-window based scheme. This is because those features are not directly supervised with language in CLIP, and moreover, these patch features have already aggregated the

global context due to self-attention, hindering the modeling of local region details for localization [23].

As for text queries, we generate several normal and abnormal descriptions. For the normal generation, the specifics are as follows, compared to industrial defect detection tasks, using textual labels to describe normal behavior under WSVAD task is more challenging. This is because videos in WSVAD task typically include multiple scenes, especially numerous real-world scenarios that are difficult to accurately summarize with textual labels directly. On the other hand, in terms of spatial fine-grained description, there may be semantic ambiguities between normal and abnormal behaviors due to the limited coverage range of spatial patches. Considering that most anomalies target intense human behavior, we believe it is more appropriate to use textual captions that describe the background of the image as normal descriptions. Therefore, we query LLMs about common indoor and outdoor items and selected 12 of the most common text descriptions as normal text descriptions. For example, "*a picture of sky, a picture of ground, a picture of road, a picture of grass, a picture of building, a picture of wall, a picture of tree, a picture of floor tile, a picture of desk, a picture of cabinet, a picture of chair, a picture of door*".

For abnormal descriptions, in addition to the original abnormal categories, we also use LLMs with a template "Provide phrases similar to [abnormal category]" to obtain augmented descriptions. For example, "*[abnormal category]*" can be set as "*people knockout someone*" for the category *Fighting*, "*people lying on the ground*" for *Car accident*, "*someone ignite fire*" for *Arson*, "*people shooting someone*" for *Shooting*. The augmented prompts, along with the original textual categories, are used as final abnormal prompts for spatial anomaly localization.

With $x_P$ and text queries $q_T$ in hands, we perform a patch-level retrieval process, namely, using normal descriptions and abnormal descriptions to locate the background regions and potential abnormal regions, respectively. Mathematically, this process can be represented as,

$$Ms[i, j] = \sum_{q_T[l] \in Anomaly} \left( \frac{exp(x_P[i, j]q_T[l]^\top / \tau)}{\sum_k exp(x_P[i, j]q_T[k]^\top / \tau)} \right) \tag{11}$$

A spatial heat map of anomalous events $Ms$ with size of $H \times W$ is created, and it is resized to the size of original frames, and can generate the predicted bounding box by shape detection algorithm. Notably, we create two different scale feature maps for $x_P$ with $P\&S$ set to 32&32 and 80&48, and use a fusion hyper-parameter $\lambda$ to average their detection results as the final result.

## 4 EXPERIMENTS

### 4.1 Datasets and Evaluation Metrics

*4.1.1 Datasets.* We conduct extensive experiments on three popular WSVAD benchmarks in which the spatio-temporal anomaly annotations of test videos are provided. **UCF-Crime** is a large-scale benchmark for WSVAD task. It consists of 1900 long and untrimmed real-world surveillance videos, where the total duration is 128 hours, and the number of training videos and test videos is 1610 and 290, respectively. **ShanghaiTech** is a medium-scale dataset of 437 videos, including 130 abnormal videos on 13 scenes.

This dataset is originally designed for semi-supervised video anomaly detection, and we follow Zhong et al. [78] and reorganize the dataset into 238 training videos and 199 test videos. **UBnormal** is a synthesized dataset. There are total 543 videos with 22 abnormal event types, in which 6 types are visible in the training set, and 12 types are visible in the test set. Following WSVAD settings, only video-level labels are available during the training stage.

*4.1.2 Evaluation metrics.* For the temporal anomaly detection, we follow previous works [52], and utilize the area under the curve (AUC) of the frame-level receiver operating characteristics (ROC). The higher AUC indicates the better performance. For the spatial anomaly localization, following the previous work [33], we use TIoU (Temporal Intersection-over-Union) as the evaluation metric, which can be formulated as the following equation:

$$TIoU = \frac{1}{N} \sum_{N}^{j=1} \frac{Area_p \cap Area_g}{Area_p \cup Area_g} \cdot I[P_j \geq Threshold], \quad (12)$$

where the indicator $I[.] \in \{0, 1\}$ indicates whether the given anomaly frame is predicted as anomaly according to the anomaly score $P_j$, $Area_p$ and $Area_g$ represent the area of prediction bounding box and the ground truth bounding box respectively, and $N$ represents the total number of all anomaly frames. We report both frame-level detection and pixel-level localization accuracy.

## 4.2 Implementation Details

We use the frozen CLIP (ViT-B/16) to extract features of video frames. Specifically, we process 1 out of 16 frames on UCF-Crime dataset, and 1 out of 4 frames on ShanghaiTech and UBnormal thus higher sampling frequency can slightly improve the performance. During the training stage, the maximum length of training videos is set to 256, videos with length exceeding the limit will be sampled to the maximum length. For all datasets, we set the length of the learnable prompts prefixed to text labels as 8. For the hyperparameters of total loss function, $\alpha$ is set to 1 on UBnormal and 0.9 on UCF-Crime and ShanghaiTech, $\beta$ is set as 2 on all datasets. $k$ is set to $[T/16] + 1$ on all datasets. For $x_{PATCH}$, we resize the image to $224 \times 224$, then use a sliding-window of size $32 \times 32$ with the stride of 32 to generate multiple patches, with both $H$ and $W$ equal to 7. Besides, $K$ in $SA^2$ is set to 12. For model optimization, we use AdamW optimizer with learning rate of 1e-4 to train the model on a single RTX3090 GPU, and batch size is set to 64.

## 4.3 Comparison with State-of-the-art Methods

To ensure fairness in comparison, we re-implement most methods using the same CLIP features as ours, given that several works utilize different feature extractors.

*4.3.1 Temporal anomaly detection results.* As listed in Tables 1 to 3, our method demonstrates superior performance on the UCF-Crime and UBnormal benchmarks, while also achieving competitive results on ShanghaiTech. Specifically, our method attains 88.08% AUC on UCF-Crime, outperforming other comparison counterparts without using CLIP features by a wide margin, and also excelling CLIP-based methods by a clear margin. Compared to the best competitor VadCLIP [71], although our STPrompt only utilizes the spatial attention aggregation instead of multi-crop augmentation,

**Table 1: Comparison of different methods on UCFCrime.**

| Method | Feature | AUC(%) | TIoU(%) |
|---|---|---|---|
| Two Stream [50] | Two-stream | 51.20 | 2.20 |
| TSN [60] | TSN | 53.20 | 2.60 |
| C3D [56] | C3D | 70.10 | 7.20 |
| T-C3D [32] | C3D | 74.50 | 10.20 |
| ARTNet [58] | ARTNets | 75.10 | 11.40 |
| 3DResNet [16] | I3D-ResNet | 77.50 | 10.30 |
| NLN [62] | I3D-ResNet | 78.90 | 12.20 |
| Liu et al. [33] | I3D-ResNet | 82.00 | 16.40 |
| SVM baseline | CLIP | 50.10 | N/A |
| OCSVM [49] | CLIP | 63.20 | N/A |
| Hasan et al. [17] | CLIP | 51.20 | N/A |
| Ju et al. [25] | CLIP | 84.72 | N/A |
| Sultani et al. [52] | CLIP | 84.14 | N/A |
| Sultani et al.[†] [52] | CLIP | 67.11 | 16.82 |
| Wu et al. [67] | CLIP | 84.57 | N/A |
| AVVD [68] | CLIP | 82.45 | N/A |
| RTFM [55] | CLIP | 85.66 | N/A |
| DMU [79] | CLIP | 86.75 | N/A |
| UMIL [39] | CLIP | 86.75 | N/A |
| CLIP-TSA [24] | CLIP | 87.58 | N/A |
| VadCLIP [71] | CLIP | 88.02 | 22.05 |
| **STPrompt** | CLIP | **88.08** | **23.90** |

**Table 2: Comparison of different methods on ShanghaiTech.**

| Method | Feature | AUC(%) | TIoU(%) |
|---|---|---|---|
| Sultani et al. [52] | CLIP | 91.72 | N/A |
| Sultani et al.[†] [52] | CLIP | 80.25 | 2.46 |
| Wu et al. [67] | CLIP | 95.24 | N/A |
| RTFM [55] | CLIP | 96.76 | N/A |
| DMU [79] | CLIP | 97.57 | N/A |
| UMIL [39] | X-CLIP[41] | 96.78 | N/A |
| MSL [30] | VideoSwin[36] | 97.20 | N/A |
| SSRL [28] | CLIP | 96.22 | N/A |
| CLIP-TSA [24] | CLIP | **98.32** | N/A |
| VadCLIP [71] | CLIP | 97.49 | 4.09 |
| **STPrompt** | CLIP | 97.81 | **9.77** |

it easily achieves a performance improvement. Besides, our method obtains 97.81% AUC, a competitive result on ShanghaiTech dataset. On UBnormal dataset, our method achieves 63.98% AUC, achieving an absolute gain of 1.0% in terms of AUC over the best competitor OPVAD [70]. The above results demonstrate the compelling ability of our method for WSVAD task, which can outperform current competition counterparts on three commonly-used benchmarks.

*4.3.2 Spatial anomaly localization results.* On the other hand, our method exhibits superior performance in spatial anomaly localization. Since there are few works exploring spatial anomaly localization, we modify a classical method [13] and an emerging method [71] as baselines, where the former is a variant of Sultani et al. [52], a simple patch-level detection, and the latter employs the

Table 3: Comparison of different methods on UBnormal.

| Method | Feature | AUC(%) | TIoU(%) |
|---|---|---|---|
| Georgescu et al. [12] | None | 59.30 | N/A |
| Georgescu et al. [12]+anomalies | None | 61.30 | N/A |
| Sultani et al. [52] | CLIP | 53.23 | N/A |
| Sultani et al.[†] [52] | CLIP | 51.95 | 5.04 |
| Wu et al. [67] | CLIP | 53.70 | N/A |
| RTFM [55] | CLIP | 60.94 | N/A |
| DMU [79] | CLIP | 59.91 | N/A |
| OPVAD [70] | CLIP | 62.94 | N/A |
| VadCLIP [71] | CLIP | 62.32 | 3.67 |
| **STPrompt** | CLIP | **63.98** | **8.17** |

learned label prompt embeddings to locate spatial anomalies. From Tables 1 to 3, we found that STPrompt achieves best performance in terms of TIoU on all the three benchmarks. For example, compared to fully-supervised method Liu et al. [33] and the baseline Sultani et al. [52], our STPrompt can achieve a substantial improvement of 7.5% and 7.2% TIoU on UCF-Crime. Besides, STPrompt also comprehensively outperforms VadCLIP, showing the advantages of purpose-built prompts created by LLMs with respect to vanilla learnable label prompts for the spatial anomaly localization.

## 4.4 Ablation Studies

To investigate the influence of designed modules, we perform extensive ablation studies with frame-level AUC and spatio-temporal-level TIoU on UCF-Crime and UBnormal benchmarks .

*4.4.1 Effectiveness of dual-branch structure.* As the result shown in Table 4, we investigate the performance of dual-branch structure in various situations. It is evident that employing both the C-branch and A-branch leads to improved performance compared to using a single branch. After adding other modules, A-branch has sustainable advantages over C-branch, which indicates that A-branch possesses superior capabilities in temporal anomaly detection, Consequently, we use the results generated from A-branch (1 minus the similarity between the video and normal class) as the final frame-level anomaly score.

*4.4.2 Effectiveness of spatial aggregation attention.* We employ the spatial aggregation attention not only for aggregating spatial features for temporal detection but also for assisting spatial anomaly location. According to the result in Table 4, using spatial features yields a notable improvement of 2.0% AUC on UCF-Crime and 1.0% AUC on UBnormal, respectively. This underscores the efficacy of spatial information in enhancing the ability of model to distinguish anomalies from normal events or background within the same frame. Consequently, abnormal regions are highlighted while redundant background is suppressed. Table 5 shows the effects of using different strategies to integrate spatial features. We observe that simply using the average features or attention-based weighted average features can slightly improve the performance on UCF-Crime, but leads to a performance drop on UBnormal. Using motion-based selection strategy can reduce redundant spatial features and achieves a improvement on both UCF-Crime and UBnormal. Our $SA^2$ is the combination of motion-based selection

Table 4: Effectiveness of each module.

| Components | | | | AUC(%) | |
|---|---|---|---|---|---|
| $SA^2$ | TemAdapter | C-Branch | A-Branch | UCF-Crime | UBnormal |
| | | ✓ | | 84.03 | 60.91 |
| | | | ✓ | 84.54 | 59.07 |
| | | ✓ | ✓ | 84.78 | 61.52 |
| ✓ | | ✓ | ✓ | 85.73 | 61.87 |
| | ✓ | ✓ | ✓ | 86.13 | 63.00 |
| ✓ | ✓ | ✓ | | 86.52 | 62.14 |
| ✓ | ✓ | | ✓ | 87.40 | 62.43 |
| ✓ | ✓ | ✓ | ✓ | **88.08** | **63.98** |

Table 5: Ablation studies on spatial modeling.

| Method | AUC(%) | |
|---|---|---|
| | UCF-Crime | UBnormal |
| w/o $SA^2$ | 86.13 | 63.00 |
| Average only | 86.91 | 62.93 |
| Attention only | 86.67 | 62.64 |
| Selection+Average | 87.79 | 63.69 |
| Selection+Attention ($SA^2$) | **88.08** | **63.98** |

and attention-based weighted average. Such a simple yet effective operation can contribute to clear improvements.

*4.4.3 Effectiveness of temporal adapter.* Previous works have proven the effectiveness of temporal relation modeling on WSVAD task. As presented in Table 4, temporal adapter gets significant performance boosts both with or without $SA^2$. Furthermore, we also perform ablation studies on how to learn temporal modeling. From Table 6, we found that the vanilla transformer that performs well when being trained on large-scale datasets with full-supervised supervision, however, is not suitable for WSVAD task. Besides, experimental results reveal that transformer based on fixed distance relation performs better than traditional self-attention transformer. This indicates that salient priors are required for WSVAD task with insufficient supervisory signals.

*4.4.4 Impact of fusion factor λ.* As aforementioned, we employ two patch-level feature maps with different size for spatial anomaly localization, which allow our model to cover spatial anomalies from different window perspectives. To further investigate the balance point between these two feature maps, we perform an ablation experiment and show the results in Table 7. For UCF-Crime, only using the feature map of size $7 \times 7$ ($\lambda$=1.0) can achieve the best performance. For another benchmark UBnormal, when two feature maps make the same contributions, i.e., setting $\lambda$ to 0.5, our model achieves the best performance. Therefore, we adopt $\lambda = 0.7$ to achieve a favorable trade-off.

## 4.5 Computational Complexity

We conduct a detailed analysis of the parameter count and computational cost of our model, juxtaposing it with previous related works in Table 8. Upon reviewing the comparison results, we note that our method, particularly when compared to spatio-temporal modeling based VAD method SSRL [28], exhibits lighter weight and

**Table 6: Ablation studies on temporal modeling.**

| Method | AUC(%) | |
| --- | --- | --- |
| | UCF-Crime | UBnormal |
| Transformer | 86.32 | 61.24 |
| Transformer+Distance Transformer | 86.97 | 61.30 |
| Distance Transformer | **88.08** | **63.98** |

**Table 7: Ablation studies on fusion factor.**

| $\lambda$ | TIoU(%) | |
| --- | --- | --- |
| | UCF-Crime | UBnormal |
| 0.0 | 20.16 | 7.31 |
| 0.3 | 19.85 | 7.96 |
| 0.5 | 21.55 | **8.18** |
| 0.7 | 23.90 | 8.17 |
| 1.0 | **24.45** | 8.14 |

**Table 8: Computational complexity comparison.**

| Method | Feature | Parameter | FLOPS |
| --- | --- | --- | --- |
| Zhong et al. [78] | C3D | 78M | 386.2G |
| RTFM [55] | I3D | 28M | 186.9G |
| SSRL [28] | I3D | 191M | 214.6G |
| SSRL* [28] | I3D | 136M | 214.6G |
| **STPrompt** | CLIP | **31.5M** | **44.8G** |

greater efficacy. It is worth highlighting that, despite the presence of shared parameters between different modules in SSRL denoted by the symbol ∗, the parameter count and computational cost of our model are notably lower. In general, the comparison results presented in Tables 1 to 3 underscore the accurate anomaly detection and localization capabilities of our method. Furthermore, these findings underscore the favorable balance between speed and accuracy achieved by our approach in spatio-temporal modeling.

### 4.6 Qualitative Analyses

*4.6.1 Qualitative results of temporal anomaly detection.* We illustrate qualitative results of temporal anomaly detection in Figure 3. The top row depicts the results of UCF-Crime, the first two samples in the bottom row present the results of ShanghaiTech, and the remaining samples show the results of UBnormal. It is clear that STPrompt can detect different types of anomalies on three public benchmarks, including human-centric anomalies, e.g., *Fighting* and *Robbery*, and scenario-centric anomalies such as *Explosion*, showcasing the effectiveness of STPrompt in anomaly detection.

*4.6.2 Qualitative results of spatial anomaly localization.* We illustrate qualitative results of spatial anomaly localization in Figure 4. For each sample, the left column is the heat map of abnormal regions, and the right column is the corresponding localization bounding boxes (red) and the ground truth bounding boxes (green). We observe that STPrompt can localize major abnormal events at the

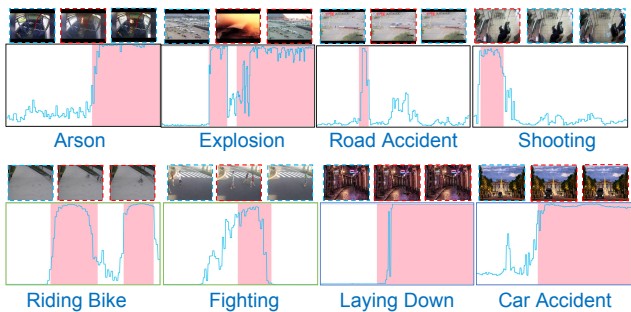

**Figure 3: Qualitative results of temporal anomaly detection on UCF-Crime, ShanghaiTech, and UBnormal.**

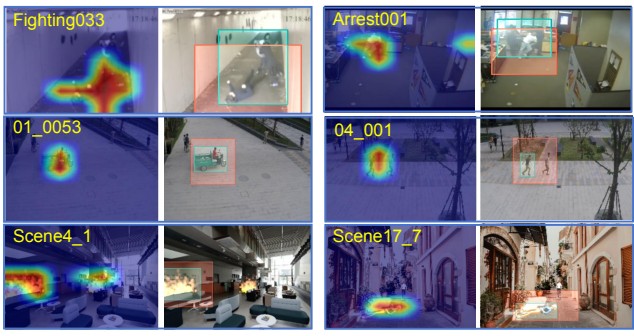

**Figure 4: Qualitative results of spatial anomaly localization on UCF-Crime, ShanghaiTech, and UBnormal.**

pixel level for these abnormal video frames, but it also produce several false alarms at the same frame. Such results demonstrate that on the one hand spatial anomaly localization is more challenging than temporal anomaly detection, and on the other, our method can better perceive anomalies at the pixel level, thereby increasing the interpretability for temporal anomaly detection.

## 5 CONCLUSION

In this work, we present STPrompt, a novel approach utilizing frozen vision-language models, for weakly supervised video anomaly detection and localization. To tackle this challenging task, we adopt a divide-and-conquer strategy by decomposing this task into two distinct sub-tasks: temporal anomaly detection and spatial anomaly localization. For the former task, we design a spatial attention aggregation strategy and temporal adapter to efficiently capture potential spatial anomaly information as well as contextual information, and then employ a dual-branch network to detect anomalies by binary classification and cross-modal alignment. For the latter task, we devise a training-free query-and-retrieve method based on the pre-trained concept knowledge from VLMs. Without bells and whistles, our STPrompt achieves state-of-the-art performance on three benchmarks in both temporal anomaly detection and spatial anomaly localization. In the future, how to further reduce the spatial false alarm rate and improve spatial localization accuracy is a problem worthy of long-standing research.

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
