# OpenReview forum: "Weakly Supervised Video Anomaly Detection and Localization with Spatio-Temporal Prompts"
_acmmm.org/ACMMM/2024/Conference — MM2024 Poster_

### Official Review · Reviewer_7GAQ · 2024-05-10

**Rating:** 5
**Confidence:** 4

**Summary:**

Authors explored weakly supervised video anomaly detection and localization task and proposed a two-stream network STPrompt to address spatio-temporal video anomaly detection under weak video-level supervisions. Authors designed a spatial attention aggregation mechanism to filter irrelevant background and provide more spatial fine-grained features for temporal anomaly detection.Inspired by CLIP-based industrial defect detection works, authors regarded the spatial anomaly localization as a spatial patch retrieval process given text queries and using LLM to generate these text prompts. Different spatial patches generate different scale feature maps, then use a fusion hyper-parameter to average their detection results as the final result.

**Strengths:**

1. The proposed work is the first endeavor to efficiently transfer pre-trained vision-language knowledge from VLMs to simultaneously tackle temporal anomaly detection and spatial anomaly localization. Overall, the novelty of this work is enough.
2. Authors proposed a novel spatial attention aggregation scheme to capture key spatial information with low computational costsuse with the motion magnitude computed by difference between frames. And the proposed method works well according to the experiment results.
3. The qualitative results of spatial anomaly localization are very intuitive and clear.

**Limitations:**

1. It seems that the improvement obtained by introducing spatial patch features is not significant on temporal anomaly detection.
2. Due to the lack of supervisions, the spatial localization still has great potential for improvement results.
3. Are these normal text descriptions good enough to describe most normal scenarios in the datasets?

**Suitability:**

3

---

### Official Review · Reviewer_LACR · 2024-05-23

**Rating:** 2
**Confidence:** 4

**Summary:**

The paper highlights that many abnormal events occur in localized regions of video frames rather than across entire frames. Existing frame-level feature-based methods may be misled by predominant background information and often lack interpretability regarding detected anomalies. To address this, a dual-stream network structure is proposed for weakly supervised video anomaly detection, with one stream focusing on the temporal dimension and the other on the spatial dimension. For temporal anomaly detection, a spatial attention aggregation mechanism is designed to help the model identify temporal anomalies more accurately. For spatial anomaly localization, a training-free strategy is proposed, utilizing fine-grained prompts provided by large language models (LLMs).

**Strengths:**

1.The motivation of the paper is clear and easily understood.

2.The idea of utilizing large language models to generate fine-grained prompts to achieve anomaly localization is interesting.

**Limitations:**

1.The methodology in the paper lacks innovation and demonstrates minimal performance enhancement. (1) The utilization of patch-level features to enhance anomaly detection is not pioneering (e.g. previous work [28]) and the design of the Temporal CLIP adapter closely resembles previous work [70].（2）The overall framework is similar to existing work [71], with improvements like adding a spatial attention mechanism and an anomaly localization branch. However, the performance improvement compared to [71] on both UCF-Crime and Shanghaitech datasets is minor. Compared with [28] which uses I3D features, this method uses CLIP features and Prompts, and the performance improvement is also small. This raises questions about the effectiveness of spatial features for anomaly detection.

2.The description of some parts of the paper is unclear. (1) In line 275, the limitations of the research work are pointed out [28, 33, 53, 64], it should be clarified whether these limitations are shared among these works, or whether each work corresponds to specific limitations. (2) The meaning of $\dag$ symbols in Table 1, Table 2 and Table 3 lacks explanation. (3) The meaning of $l$ in Equation (11) should be clarified. And $l$ is also used in line 456, which is confusing. (4) Please indicate how many augmented descriptions were generated using LLM, and explain how these descriptions were merged with the original text categories to form the final prompt.

3.In lines 611-613, the sampling method used in the training process appears to differ from the previous methods. Does this change have a significant impact on performance? Please provide details of the specific sampling method used.

4.Among the qualitative results shown in Figure 4, whether the anomaly localization results can correspond to the prompt description.

5.How about the performance of the proposed method on XD-Violence dataset?

**Suitability:**

3

---

### Official Review · Reviewer_AvEj · 2024-05-25

**Rating:** 4
**Confidence:** 3

**Summary:**

This paper proposes a spatial-temporal prompt embedding for weakly supervised video anomaly detection using pretrained vision language models (VLMs). Experiments show comparative performance with reduced computational complexity. The paper is generally well-written and easy to follow.

**Strengths:**

The use of LLM to generate augmented descriptions for spatial-temporal anomaly localization.
There are extensive baselines in experiments, and the experiments cover multiple aspects of performance.

**Limitations:**

Three sophisticated datasets are used, which is a good thing, and it would be better if per-dataset comparisons are made against the baselines.

**Suitability:**

2

---

### Official Review · Reviewer_kuse · 2024-06-02

**Rating:** 4
**Confidence:** 3

**Summary:**

The paper introduces a novel method that learns spatio-temporal prompt embeddings for weakly supervised video anomaly spatio-temporal localization based on pre-trained vision-language models (VLM).
The proposed method employs two-stream network structure, one for temporal anomaly detection and the other for spatial anomaly localization.
For the temporal anomaly detection branch, a spatial attention aggregation (SA2) scheme is proposed to capture key spatial information.
For spatial anomaly localization branch, some textual captions of background (normal descriptions) and some textual captions of abnormal descriptions are added as augmented text queries.

**Strengths:**

The idea of video anomaly spatio-temporal localization without object detection/tracking is novel.
The proposed method achieves state-of-the-art performance on three public benchmarks for the WSVADL task.

**Limitations:**

(1) Some symbols are not explained, such as ‘𝑀𝑜𝑇𝑂𝑃 [𝑖] ’ in Eq.(2).
(2) In Table 1, the performance of the proposed method that has an extra spatial anomaly localization branch is a little better than those of VadCLIP (88.08 vs. 88.02). Does it mean that spatial anomaly localization do not bring much benefit for video anomaly detection?
(3) One advantage of the proposed method is lighter weight. I’m curious about How it is compared with those of VadCLIP.

**Suitability:**

3

---

### Meta-Review · Area_Chair_QyV9 · 2024-07-02

**Recommendation:** Accept (Poster)
**Confidence:** 5

**Metareview:**

The initial ratings are 1 WR, 2 BA and 1 WA, and they are 1 BA, 1 WA and 1 Accept after rebuttal. The reviewers all appreciated the proposed method about transfer pre-trained vision-language knowledge from VLMs to simultaneously tackle temporal anomaly detection and spatial anomaly localization, and are satisfied with the authors' responses.
I agree with the reviewers and recommend an acceptance. The authors are encouraged to incorporate the material from their response to the camera ready version.